# Dexamethasone Suppression Testing in a Contemporary Cohort with Adrenal Incidentalomas in Two U.S. Integrated Healthcare Systems

**DOI:** 10.3390/biomedicines11123167

**Published:** 2023-11-28

**Authors:** Mackenzie Crawford, Bennett McDonald, Wansu Chen, Hina Chowdhry, Richard Contreras, Iris Anne C. Reyes, Eleena Dhakal, Tish Villanueva, Joshua I. Barzilay, Candace F. Vaughn, Frank S. Czerwiec, David A. Katz, Annette L. Adams, Jennifer C. Gander

**Affiliations:** 1Center for Research and Evaluation, Kaiser Permanente Georgia, Atlanta, GA 30305, USA; 2Department of Research and Evaluation, Kaiser Permanente Southern California, Pasadena, CA 91101, USA; 3Southeastern Permanente Medical Group, Kaiser Permanente Georgia, Atlanta, GA 30305, USA; 4Sparrow Pharmaceuticals, Portland, OR 97204, USA

**Keywords:** autonomous cortisol secretion, adrenal incidentaloma, dexamethasone suppression test

## Abstract

Autonomous cortisol secretion (ACS) from an adrenal adenoma can increase the risk for comorbidities and mortality. The dexamethasone suppression test (DST) is the standard method to diagnose ACS. A multi-site, retrospective cohort of adults with diagnosed adrenal tumors was used to understand patient characteristics associated with DST completion and ACS. Time to DST completion was defined using the lab value and result date; follow-up time was from the adrenal adenoma diagnosis to the time of completion or censoring. ACS was defined by a DST > 1.8 µg/dL (50 nmol/L). The Cox proportional hazards regression model assessed associations between DST completion and patient characteristics. In patients completing a DST, a logistic regression model evaluated relationships between elevated ACS and covariates. We included 24,259 adults, with a mean age of 63.1 years, 48.1% obese, and 28.7% with a Charlson comorbidity index ≥ 4. Approximately 7% (n = 1768) completed a DST with a completion rate of 2.36 (95% CI 2.35, 2.37) per 100 person-years. Fully adjusted models reported that male sex and an increased Charlson comorbidity index were associated with a lower likelihood of DST completion. Current or former smoking status and an increased Charlson comorbidity index had higher odds of a DST > 1.8 μg/dL. In conclusion, clinical policies are needed to improve DST completion and the management of adrenal adenomas.

## 1. Introduction

Autonomous cortisol secretion (ACS) is a condition involving excess secretion of cortisol from an adrenal adenoma [1]. While most adrenal adenomas are nonfunctioning, recognition that many adenomas autonomously secrete cortisol has increased [2]. In addition, more adrenal adenomas have been identified incidentally with the increased use of diagnostic imaging over the last two decades [1,2,3]. Also known as adrenal incidentalomas, they are defined as a clinically unapparent and benign-appearing adrenal mass greater than one centimeter in diameter detected during imaging performed for reasons unrelated to suspicion of primary or metastatic adrenal disease [4,5,6,7]. They are most often found in adults aged 65 or older, with previous studies showing prevalence rates varying from 5% to 7% among that age group [3].

Non-malignant adrenal adenomas can be divided into those that are and are not hormonally active. Recent studies suggest that approximately 30% of adrenal adenomas are hormonally active, though the optimal hormonal workup upon adrenal adenomas diagnosis is frequently not completed [2]. In patients with hormonally active adrenal adenomas, ACS is diagnosed based on morning cortisol >1.8 µg/dL (50 nmol/L) in a 1 mg overnight dexamethasone suppression test (DST) [5,6,7]. ACS has been found to be associated with increased cardiovascular morbidity and metabolic abnormalities, as well as increased mortality [8,9]. All-cause mortality was significantly increased in patients with ACS, both those with a DST result of >1.8–5.0 µg/dL and those with a DST result >5.0 µg/dL. A higher prevalence of hypertension in patients with either level of ACS was found compared to those with nonfunctioning adrenal tumors.

Establishing and adhering to a diagnostic workflow is invaluable to addressing the increases in morbidity and mortality found in ACS patients, though not without challenges. The 2016 and 2023 European Society of Endocrinology guidelines recommended applying a cut-off value of DST serum cortisol ≤1.8 µg/dL as an exclusionary ACS [5,7]. In those guidelines, patients with serum cortisol levels >5.0 µg/dL but without showing any clinical signs of Cushing’s syndrome were classified as having ACS, and those with levels 1.9–5.0 µg/dL were classified as having possible ACS. This guidance has been updated in 2023 to define ACS based on serum cortisol levels of >1.8 µg/dL without any further stratification based on the degree of cortisol non-suppressibility [5,7]. In such patients, replication of DST results and further hormonal workup to confirm ACTH independence are now also recommended.

While the recommended workflow to identify whether the adrenal adenoma was secreting cortisol relied on DST completion since 2012, evidence suggests DST is frequently not completed in clinical settings, although the circumstances are not clear [2,10]. Additionally, while the literature reports the increased risk for mortality and comorbidities for individuals with ACS, most studies examined a small number of participants and were unable to determine the patient-level factors associated with ACS. We created a diverse retrospective cohort from two integrated health systems to analyze the diagnostic workflow for patients who have adrenal adenomas and, potentially, ACS. Secondly, among those in the cohort who did have a DST, we examined the associations of patient characteristics with the different ranges of DST results. We first aimed to better understand the risk factors associated with completing a DST and then to assess the characteristics associated with ACS. Our goal is to further guide the physician workflow for patients who might have ACS.

## 2. Methods

### 2.1. Population

This study was conducted at two integrated Kaiser Permanente healthcare systems in Georgia and Southern California. Kaiser Permanente of Georgia (KPGA) encompasses the metro Atlanta area and surrounding counties (>300,000 active members), and Kaiser Permanente of Southern California (KPSC) encompasses a broad range of the Southern California population (>4.8 million active members).

The eligible members for this study were individuals with evidence of potential incident ACS between 2015 and 2022. Participants with an adrenal adenoma between 1 January 2015 and 31 August 2022 were identified using either ICD-9 or ICD-10 diagnosis codes (KPSC and KPGA) or problem list items indicating adrenal adenoma (KPGA only) (Appendix A). Diagnosis codes and problem list items were recommended and reviewed by endocrinologists at each site. The date of the earliest adrenal adenoma diagnosis or documentation in the problem list was designated as their adrenal adenoma index date. Individuals 18 years of age or older at their index date and enrolled with the health system for at least one day during the study period were included. Those with any evidence of Cushing’s disease diagnosis or hyperadrenalism (n = 831), malignant adrenal cancer (n = 416), adrenalectomy (n = 1218), or potential ACS prior to 2015 (n = 13,543) were excluded from the cohort in that order (Figure 1). Institutional Review Board approval was obtained at each site: Kaiser Permanente Interregional Institutional Review Board for KPGA (IRB# 2010058) and KPSC Institutional Review Board (IRB# 13485).

### 2.2. Outcomes of Interest

Two primary outcomes were assessed for this study. First, the relationship between member characteristics and the likelihood of completing a DST was assessed. DST completion was defined using the DST lab value and result date and was dichotomized as either complete or not complete. Participants were followed prospectively from the member index date (date of first documented adrenal adenoma) until the study end date on 31 August 2023. Time from adrenal adenoma diagnosis to event or censoring was measured as the number of days from the ACS index date until completion of DST (primary event), disenrollment from the health system, death, or study end date. Our second outcome of interest was DST values among individuals with a completed DST. We used the first DST result after the adrenal adenoma diagnosis date. We categorized the DST value into two groups based on clinical guidelines for ACS (≤1.8 µg/dL, >1.8 µg/dL).

### 2.3. Covariates

Our goal was to assess the patient characteristics associated with DST completion and a DST value >1.8 µg/dL. Covariate data were derived from a combination of health system medical records, billings, and claims data and included demographic, behavioral, clinical, and neighborhood characteristics at the time of adrenal adenoma diagnosis with a maximum one-year look-back period. Age was categorized based on risk strata (18–39, 40–49, 50–59, 60–69, and 70+). Member sex and race/ethnicity were derived from medical record data. Neighborhood characteristics were captured using a census tract-level neighborhood deprivation index derived from the American Community Survey, a standardized score ranging from −3 to 3, with higher scores indicating greater neighborhood deprivation [11]. Deprivation scores were categorized into quintiles, with the last quintile (Q5) representing the highest and most deprived neighborhoods. A history of pre-diabetes, diabetes, hypertension, chronic kidney disease stage 3–5, dyslipidemia, coronary artery disease, myocardial infarction, peripheral vascular disease, cerebral vascular disease, stroke, transient ischemic attack, deep vein thrombosis, osteoporosis, and osteopenia was obtained from the electronic medical record and categorized as ‘yes’ if the individual ever had a diagnosis for the specific comorbidity prior to the adrenal adenoma date. Cardiovascular disease was a composite binary variable defined as the presence of coronary artery disease, myocardial infarction, or peripheral vascular disease. The Charlson Comorbidity Index score included a history of chronic comorbidities [12], was calculated from a 5-year medical history prior to the adrenal adenoma diagnosis date, and was categorized into four categories (0–1 comorbidities, 2 comorbidities, 3 comorbidities, and 4+ comorbidities).

Other clinical and behavioral variables were assessed from healthcare encounters within the 365 days prior to and 90 days after the ACS index date. Height and weight measures were used to calculate body mass index (BMI) from the encounter closest to the adrenal adenoma date. Smoking status and smoking history are routinely measured during healthcare encounters via survey; the survey response closest to the member’s adrenal adenoma diagnosis date was used to categorize members into current or former smokers vs. never smokers. Lab values were captured, and the result closest to the index date was recorded for the following labs: HbA1c, creatinine, total cholesterol, LDL, HDL, triglycerides, and potassium.

### 2.4. Descriptive Analysis

Patient descriptive characteristics at the time of adrenal adenoma diagnosis were described overall and compared across sites using chi-squared tests and *t*-tests. Average follow-up time, number of completed DSTs, and DST completion rate per 100 person-years were described by demographic, behavioral, and clinical groups. Member characteristics were described by the two DST result categories overall and by site.

### 2.5. Time to DST Completion

Cox proportional hazard models were used to assess the association between the time to DST completion and demographic, behavioral, and clinical characteristics. We tested the proportional hazards assumption for each covariate by examining log-log survival curves for parallel patterns of the survival function across time. In addition, we built a model regressing each covariate factor interacting with time to test for non-proportionality. We used both results to determine whether a covariate met the proportional hazards assumption.

Unadjusted models containing each covariate factor were built, and hazard ratios with 95% confidence intervals describing the association with time to DST completions were reported. For covariates that did not meet the proportional hazards assumption, we included an interaction term with time in all unadjusted and adjusted models. Results without interaction are reported in descriptive results to provide an overview of the overall covariate effect.

For adjusted models, we used a nested model approach with conceptually related covariate groupings and backward elimination to assess model fit. The covariates were grouped based on Andersen’s behavioral model of health services use [13,14]; models were built using a nesting approach with Model 1, including adrenal adenoma diagnosis year. Model 2 included Model 1 and considered sociodemographic characteristics (age, sex, race/ethnicity) and the neighborhood deprivation index; Model 3 considered behavioral characteristics such as smoking and obesity; and Model 4 considered individual comorbid conditions such as diabetes, hypertension, and the Charlson Comorbidity Index. Likelihood ratio tests and the Akaike information criterion were used to compare nested models until a best-fit model was decided upon. It was determined a priori that the ACS diagnosis year and sociodemographic characteristics would be retained in all models.

### 2.6. Elevated DST Result

Logistic regression was used to assess the association between DST normal (≤1.8 µg/dL) and elevated (DST > 1.8 µg/dL) and sociodemographic, behavioral, and clinical characteristics. For unadjusted analysis, each covariate was fit individually. For the adjusted models, a subset of the covariates that were associated with DST resulted in bivariate analyses and were determined to have clinical significance were selected. Using the same conceptual framework described above, a nested model approach with conceptually related variable groupings and backward selection was built using the same approach described above for the DST completion analysis. All analysis was performed using SAS Enterprise Guide version 8.2.

## 3. Results

A total of 24,259 patients across the two study sites (Figure 1) were found to have evidence of an adrenal adenoma and were included in the retrospective study. KPSC included 21,949, and KPGA included 2310 (Table 1). The mean age at the time of adrenal adenoma diagnosis was 63.1 years, with KPGA having a slightly younger age than KPSC (60.9 vs. 63.4 years, *p*-value < 0.0001). Overall, 58.2% of the cohort were female, similar to the stratified cohorts (*p*-value = 0.29). The two sites had a similar proportion of non-Hispanic White individuals, which averaged 43.9% overall. KPGA had a higher percentage of Black or African–American (47.5%) individuals compared to KPSC (12.0%); KPSC has a higher percentage of Hispanic individuals (32.4%) compared to KPGA (3.0%). The mean Neighborhood Deprivation Index (NDI) was reported in quintiles, with the highest score seen for Q5 in the overall cohort (1.57) as well as in KPGA (1.24) and KPSC (1.60). KPGA had more participants who were current or past smokers (51.4%) compared to KPSC (48.2%, *p*-value = 0.0031). The average HbA1c at the time of adrenal adenoma diagnosis was 6.4 (±1.3) mmol/mol in the total cohort and similar across KPGA and KPSC. The average LDL at the time of adrenal adenoma diagnosis was higher for KPGA members (109.2 ± 40.1 mg/dL) compared to KPSC members (99.0 ± 38.9 mg/dL, *p*-value < 0.0001). A total of 46.9% of the cohort had a Charlson Comorbidity Index score of 0–1, 28.7% had a score of 4 or higher, and the remaining 24.4% scored a 2–3.

Table 2 reports the average follow-up time (3.12 years), number of DST tests performed (1768), and rate of performed DST per 100 person-years (IR = 2.36, 95% CI 2.35, 2.37) among all individuals with an adrenal adenoma. Members over the age of 70 (IR = 1.80, 95% CI 1.80, 1.81), males (IR = 2.04, 95% CI 2.03, 2.05), non-Hispanic White (IR = 1.92, 95% CI 1.92, 1.93), and multiracial (IR = 1.93, 95% CI 1.80, 2.18) had lower rates of DST completion compared to their counterparts. Members with a diagnosis of chronic kidney disease stage 3–5 (IR = 1.90, 95% CI 1.89, 1.93), coronary artery disease (IR = 1.75, 95% CI 1.74, 1.78), myocardial infarction (IR = 1.50, 95% CI 1.47, 1.55), and transient ischemic attack (IR = 1.65, 95% CI 1.59, 1.77) had DST completion rates below the overall rate. Those with a Charlson comorbidity score ≥ 4 had lower rates of DST completion (IR = 1.88, 95% C 1.87, 1.89) compared to individuals with a Charlson comorbidity index 0–1 (IR = 2.57, 95% CI 2.57, 2.58). The rates of DST completion increased monotonically by adrenal adenoma diagnosis year, from 0.41 (95% CI 0.41, 0.42) in 2015 to 13.65 (95% CI 13.54, 13.85) in 2022.

The associations between patient characteristics and time to DST completion are reported in Table 3. The crude HR decreased as patient age increased, from 1.10 (95% CI 0.89, 1.38) for those aged 40–49 years to 0.64 (95% CI 0.52, 0.78) for those aged ≥ 70 years. Male participants had a crude HR of 0.79 (95% CI 0.72, 0.87). Black or African–American individuals were more likely to complete a DST, with a crude HR of 1.46 (95% CI 1.28, 1.67) compared to non-Hispanic White individuals. Patients with obesity (BMI ≥ 30 kg/m^2^) had a crude HR of 1.26 (95% CI 1.15, 1.39) compared to those individuals with a BMI < 30 kg/m^2^. Similar patterns were seen in the fully adjusted time-to-event model. Participants aged ≥ 70 years (HR = 0.84, 95% CI 0.67, 1.05), males (HR = 0.84, 95% CI 0.76, 0.93), and individuals with a Charlson comorbidity index score ≥ 4 (HR = 0.80, 95% CI 0.67, 0.96) were less likely to complete a DST compared to their counterparts. Black and African–American members were more likely to complete a DST (HR = 1.27, 95% CI 1.10, 1.46) compared to non-Hispanic white members. The likelihood of completing a DST was higher for individuals with increased NDI, with Q5 having the highest at 1.33 (95% CI 1.13, 1.57) compared to Q1. Individuals with an adrenal adenoma diagnosed in 2020 (HR = 26.71, 95% CI 16.86, 42.34), 2021 (HR = 41.45, 95% CI 25.65, 66.97), and 2022 (HR = 54.03, 95% CI 33.01, 88.43) were significantly more likely to complete a DST compared to the 2015 diagnosis year.

Among the 24,259 patients diagnosed with an adrenal adenoma, 1768 (7.4%) completed a DST during the study period (Table 4). A total of 665 (37.6%) had DST results of >1.8 μg/dL. The average age of those whose DST result was ≤1.8 µg/dL was 59.2 years, while that of those whose test result was >1.8 µg/dL was 62.4 years. Similar results were seen across study sites, with KPGA showing a slightly younger demographic for both test result categories. Of the total population with a completed DST result ≤ 1.8 µg/dL, 63.1% were females. In the total cohort with a DST result > 1.8 μg/dL, 22.4% were Black or African–American individuals, 28.6% were Hispanic individuals, 49.2% were current or past smokers, 50.2% were obese, 43.5% had a Charlson comorbidity index score 0–1, and 23.6% had an adrenal adenoma diagnosis in 2022. Similar percentages were seen for site-specific characteristics.

The crude odds ratios of having DST > 1.8 μg/dL are reported in Table 5. The crude OR was found to increase with each increasing age category compared to the referent group of 18–49, with the highest seen among individuals aged≥ 70 (OR = 2.80, 95% CI 1.76, 4.48). For the remaining demographics, the highest crude odds ratios, compared to the referent group of non-Hispanic Whites, were found in Black or African–American members (OR = 1.29, 95% CI 0.99, 1.70), patients in the Q2 of the neighborhood deprivation index quartiles (OR = 1.23, 95% CI 0.89, 1.71) compared to the referent group of Q1, current or past smokers (OR = 1.39, 95% CI 1.15, 1.69) compared to the referent group of those who had never smoked, and individuals with Charlson comorbidity index ≥ 4 (OR = 2.77, 95% CI 2.16, 3.56) compared to the referent group of 0–1. Individuals with an adrenal adenoma diagnosis in 2017 were (OR = 2.06, 95% CI 1.03, 4.13) more likely to have a DST > 1.8 μg/dL compared to individuals diagnosed in 2015. After the model was adjusted for patient demographics, behavioral, and clinical characteristics (Figure 2), members aged 50–59 years had higher odds of DST > 1.8 μg/dL (OR = 1.47, 95% CI 0.90, 2.40) compared to members aged 18–39 years. Black or African–American individuals had 1.28 (95% CI 0.97, 1.70) times the odds of having had a DST > 1.8 μg/dL compared to non-Hispanic White individuals. Adults who are current or former smokers and individuals with a Charlson comorbidity index ≥ 4 were 1.23 (95% CI 1.00, 1.51) and 2.71 (95% CI 205, 3.58) times at higher odds of having had a DST > 1.8 μg/dL.

## 4. Discussion

An incidental finding of an adrenal adenoma, also known as an adrenal incidentaloma, is becoming more common with increased imaging use and technology [5,6,15]. Guidelines for the evaluation of adrenal adenomas include the completion of a DST and the management of a functioning adenoma based on a DST > 1.8 μg/dL [6,16,17]. We aimed to better understand the implementation of these guidelines, describe the DST completion rate, and investigate the patient characteristics associated with a DST > 1.8 μg/dL across two large integrated healthcare systems. Among 24,259 adults with an adrenal adenoma diagnosed between 2015 and 2022, 1768 (7.29%) completed a DST, of which 37.61% had a DST > 1.8 μg/dL. Males and individuals with a higher Charlson comorbidity index were less likely to complete a DST. Individuals with an adrenal adenoma diagnosis in more recent years were more likely to complete a DST compared to 2015. Patient characteristics associated with an increased likelihood of having a DST > 1.8 μg/dL included increased age, being a current or former smoker, and a higher Charlson comorbidity index.

Kapoor et al. published a 2011 report on the guidelines for detecting and managing adrenal adenomas [17]. They state that the literature supports a DST with reported high sensitivity and specificity and provides principles for adrenal incidentaloma evaluation [18,19]. In multiple guideline reports, authors encourage all adrenal incidentalomas to undergo thorough clinical and hormonal testing to distinguish between hyperfunctioning and benign masses [1,5,17,20]. However, despite these recommendations from >10 years prior and additional guidelines presented in the European Network for the Study of Adrenal Tumors [5], we found only a small percentage of our members with a completed DST since 2015, although this proportion of completed DST did increase from 3.11% in 2015 to 27.48% in 2022. We noticed a 142% increase in DST completion from 2019 to 2021 that may be reflective of a standardized workflow KPGA established around incidentalomas in 2020 and specifically for adrenal incidentalomas in 2021. Despite the increase in DST completion, the increase in completion is still modest and may be considered clinical inertia based on the lack of management and testing for the adrenal adenoma [21]. Clinical inertia increases the chance of adverse events or worsening outcomes. We urge healthcare systems to standardize adrenal adenoma management workflows and best practices.

Autonomous cortisol secretion has been associated with an increased risk of diabetes, cardiovascular disease, and mortality [3,8,22,23,24,25,26,27]. While more of the literature reports on the outcomes of functioning adrenal incidentalomas with DST > 1.8 μg/dL, the literature is sparse on the patient characteristics associated with a functioning adrenal adenoma. While there is guidance around the appropriate use of DST [28] and clinical considerations for ACS diagnosis [29], few reports discuss patient characteristics that may increase a patient’s likelihood of having a functioning adrenal adenoma. In a small, single-center retrospective cohort study of 197 individuals, the authors found 76 (38.6%) of patients had a DST > 1.8 μg/dL [30]. Consistent with previous reports, our study found that, among the individuals with a completed DST, 37.61% had a DST > 1.8 μg/dL. In adjusted regression analyses, we determined that current or former smoking status and an increased Charlson comorbidity index increased the risk for a DST > 1.8 μg/dL, while male sex and obesity (BMI ≥ 30 kg/m^2^) seemed to reduce the risk for a DST > 1.8 μg/dL. Larger, retrospective studies are needed to better understand the risk factors associated with autonomous cortisol secretion, implement more upstream approaches, and increase the prevention of poor outcomes and mortality.

Our study combined two geographically and demographically diverse integrated healthcare populations with robust electronic medical records and longitudinal follow-up. However, our study is not without limitations. First, during the development of cohorts, we recognized the differences in electronic medical record documentation and the capture of adrenal adenoma. KPGA relied primarily on problem-list documentation, and KPSC utilized diagnosis codes. While both sites implemented both the problem list and diagnosis codes as part of their adrenal adenoma definition, we may have missed cases and underestimated the DST completion rate. Second, although the lab values may have provided meaningful insight into risk factors associated with autonomous cortisol secretion, there was a high volume of missing lab value data (for HbA1C, cholesterol panel, and potassium) around the time of adrenal adenoma diagnosis, and these covariates could not be included in the model. Standardized workflows should be established to ascertain a complete metabolic and blood panel in patients with adrenal adenoma diagnoses. Third, we acknowledge that DST may have diminished accuracy because the test requires the patient to follow strict instructions, and non-compliance may alter the results. Additional investigation on larger cohorts is needed to reproduce these analyses using a larger population of DST-completed results to reduce this inherited variance.

In conclusion, we found that among a cohort of 24,259 adults, fewer than 10% of individuals with an incident adrenal adenoma completed a DST between 2015 and 2022, and more than one in three individuals with a completed DST had a functioning adrenal adenoma (DST > 1.8 μg/dL). Although utilization of DST has increased dramatically since 2019, more work needs to be conducted to create standardized management plans for adrenal incidentalomas. Researchers need to also continue to investigate the patient characteristics associated with autonomous cortisol secretion to create improved screening protocols and reduce the risk of poor outcomes and mortality.

## Figures and Tables

**Figure 1 biomedicines-11-03167-f001:**
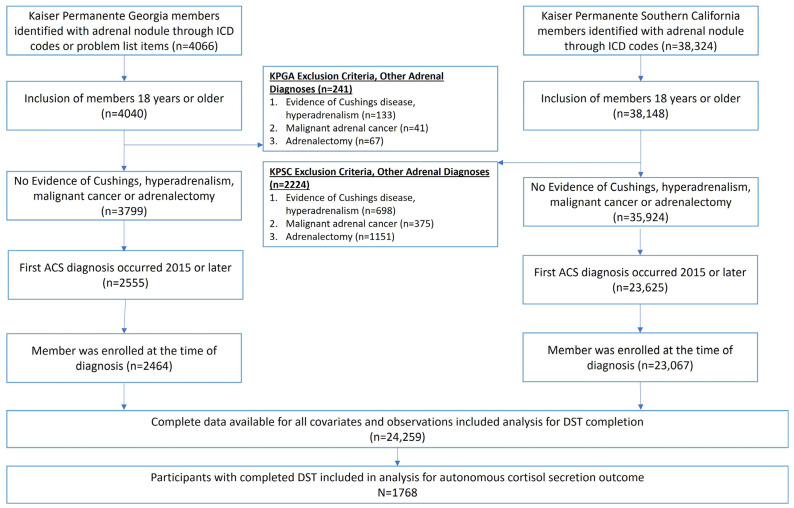
Flow diagram of multi-site retrospective cohort creation among all adults with diagnosed adrenal incidentaloma, 2015–2022.

**Figure 2 biomedicines-11-03167-f002:**
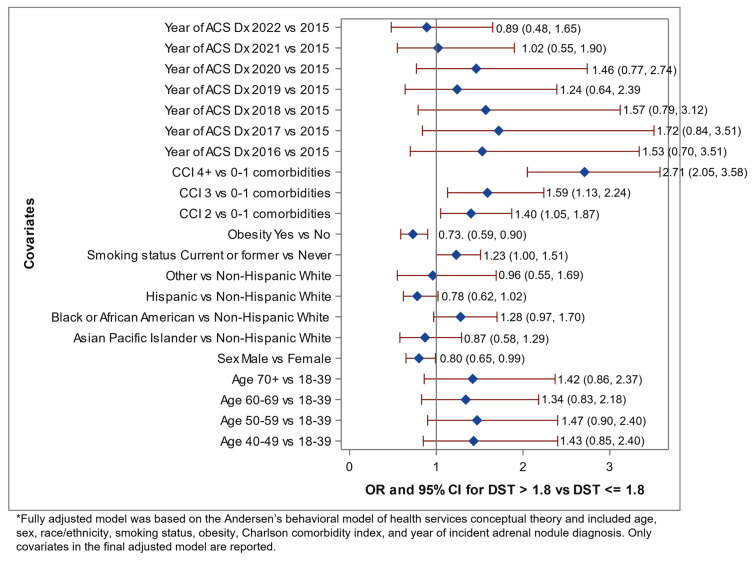
Forest plot reporting the final, fully adjusted* odd ratios (OR) and 95% confidence intervals (95% CI) assessing the association between autonomous cortisol secretion (ACS), defined as DST > 1.8 mg/dL, for all adults with completed dexamethasone suppression test (DST), 2015–2022.

**Table 1 biomedicines-11-03167-t001:** Patient demographics and clinical characteristics among all adults with adrenal adenoma for the total population and stratified by healthcare region, 2015–2022.

	Total Population N (%)	KP Georgia N (%)	KP Southern California N (%)	Comparison Test *p*-Value
Patient Characteristics at the Time of Adenoma Diagnosis	N = 24,259	N = 2310 (9.52)	N = 21,949 (90.48)	
Age, mean (SD)	63.1 (13.9)	60.9 (12.6)	63.4 (14.0)	<0.0001
Age group, n (%)				<0.0001
18–39	1443 (5.9)	106 (4.6)	1337 (6.1)	
40–49	2533 (10.4)	317 (13.7)	2215 (10.1)	
50–59	4893 (20.2)	597 (25.8)	4293 (19.6)	
60–69	7134 (29.4)	722 (31.3)	6408 (29.2)	
70+	8274 (34.1)	568 (24.6)	7696 (35.1)	
Sex, n (%)				0.29
Male	10,151 (41.8)	942 (40.8)	9198 (41.9)	
Female	14,126 (58.2)	1368 (59.2)	12,751 (58.1)	
Patient Race/Ethnicity, n (%)				<0.0001
Asian Pacific Islander	1890 (7.8)	73 (3.2)	1821 (8.3)	
Black or African–American	3727 (15.4)	1097 (47.5)	2631 (12.0)	
Non-Hispanic White	10,653 (43.9)	980 (42.4)	9663 (44.0)	
Hispanic	7179 (29.6)	69 (3.0)	7105 (32.4)	
Multiracial	316 (1.3)	10 (0.4)	306 (1.4)	
Other	48 (0.2)	5 (0.2)	43 (0.2)	
Unknown	464 (1.9)	76 (3.3)	380 (1.7)	
Neighborhood Deprivation Index (NDI), Mean (SD)				<0.0001
Q1	−0.92 (0.2)	−1.08 (0.3)	−0.90 (0.2)	
Q2	−0.39 (0.1)	−0.49 (0.1)	−0.38 (0.1)	
Q3	0.06 (0.1)	−0.07 (0.1)	0.08 (0.1)	
Q4	0.60 (0.2)	0.37 (0.2)	0.63 (0.2)	
Q5	1.57 (0.6)	1.24 (0.6)	1.60 (0.5)	
Smoking status, n (%)				0.0031
Current or Past	11,764 (48.5)	1187 (51.4)	10,568 (48.2)	
Never smoked	12,513 (51.5)	1123 (48.6)	11,381 (51.9)	
Lab Values, mean ± SD (median)				
HbA1c	6.4 ± 1.3 (6.0)	6.3 ± 1.4 (5.9)	6.4 ± 1.3 (6.0)	0.0070
Creatinine	1.0 ± 0.9 (0.9)	1.1 ± 1.2 (0.9)	1.0 ± 0.7 (0.9)	0.0002
Total cholesterol	172.3 ± 46.7 (169.0)	181.3 ± 45.5 (178.0)	171.4 ± 46.7 (168.0)	<0.0001
LDL	100.1 ± 39.2 (95.0)	109.2 ± 40.1 (107.0)	99.0 ± 38.9 (94.0)	<0.0001
HDL	49.4 ± 14.7 (47.0)	51.3 ± 15.3 (49.5)	49.2 ± 14.6 (47.0)	<0.0001
Triglycerides	168.7 ± 200.8 (137.0)	130.8 ± 100.4 (109.0)	173.7 ± 210.2 (141.0)	<0.0001
Potassium	4.0 ± 0.5 (4.0)	4.0 ± 0.5 (4.0)	4.0 ±0.5 (4.0)	<0.0001
Comorbidities, n (%)				
Obesity	11,680 (48.1)	1205 (52.2)	10,466 (47.7)	<0.0001
Pre-diabetes	6859 (28.3)	617 (26.7)	6239 (28.4)	0.08
Diabetes	9036 (37.2)	811 (35.1)	8214 (37.4)	0.03
Hypertension	16,317 (67.2)	1671 (72.3)	14,630 (66.7)	<0.0001
Chronic Kidney Disease, Stage 3–5	4135 (17.0)	311 (13.5)	3812 (17.4)	<0.0001
Dyslipidemia	15,321 (63.1)	1022 (44.2)	14,284 (65.1)	<0.0001
Cardiovascular Disease	17,522 (72.2)	1741 (75.4)	15,765 (71.8)	0.0003
Coronary Artery Disease	3380 (13.9)	321 (13.9)	3056 (13.9)	0.97
Myocardial Infarction	1669 (6.9)	154 (‘6.7)	1514 (6.9)	0.73
Peripheral Vascular Disease	6862 (28.3)	464 (20.1)	6391 (29.1)	<0.0001
Cerebral Vascular Disease	2457 (10.1)	255 (11.0)	2202 (10.0)	0.13
Deep Vein Thrombosis	244 (1.0)	28 (1.2)	215 (1.0)	0.27
Transient Ischemic Attack	760 (3.1)	80 (3.5)	680 (3.1)	0.35
Stroke	1005 (4.1)	118 (5.1)	887 (4.0)	0.02
Osteoporosis	2958 (12.2)	150 6.5)	2807 (12.8)	<0.0001
Osteopenia	3428 (14.1)	206 (8.9)	3222 (14.7)	<0.0001
Charlson Comorbidity Index				<0.0001
0–1	11,388 (46.9)	1192 (51.6)	10,194 (46.4)	
2	3581 (14.8)	320 (13.9)	3259 (14.8)	
3	2339 (9.6)	224 (9.7)	2112 (9.6)	
4+	6969 (28.7)	574 (24.9)	6384 (29.1)	
Year of incident adrenal adenoma diagnosis n (%)				<0.0001
2015	2513 (10.4)	212 (9.2)	2299 (10.5)	
2016	2662 (11.0)	269 (11.7)	2391 (10.9)	
2017	2921 (12.0)	301 (13.0)	2617 (11.9)	
2018	3019 (12.4)	288 (12.5)	2730 (12.4)	
2019	3195 (13.2)	278 (12.0)	2913 (13.3)	
2020	2867 (11.8)	211 (9.1)	2656 (12.1)	
2021	3569 (14.3)	320 (13.9)	3149 (14.3)	
2022	3631 (15.0)	431 (18.7)	3194 (14.6)	

**Table 2 biomedicines-11-03167-t002:** Among all individuals with an adrenal adenoma diagnosis, the average follow-up time, the number of completed dexamethasone suppression tests (DSTs), and the rate of complete DST after the first adrenal adenoma.

Patient Characteristics at the Time of Adenoma Diagnosis	Mean Follow Up in Years (SD)	Number with Completed DST	Rate of Completed DST (95% CI) per 100 PY
Overall	3.12 (2.43)	1768	2.36 (2.35, 2.37)
Age group, n (%)			
18–39	2.78 (2.29)	119	2.97 (2.93, 3.06)
40–49	3.01 (2.37)	238	3.12 (3.10, 3.16)
50–59	3.28 (2.45)	411	2.56 (2.55, 2.58)
60–69	3.29 (2.46)	557	2.38 (2.37, 2.39)
70+	2.97 (2.40)	443	1.80 (1.80, 1.81)
Sex, n (%)			
Male	3.14 (2.42)	638	2.04 (2.03, 2.05)
Female	3.09 (2.44)	1130	2.55 (2.54, 2.55)
Patient Race/Ethnicity, n (%)			
Asian Pacific Islander	3.31 (2.47)	139	2.21 (2.19, 2.26)
Black or African–American	3.09 (2.49)	324	2.81 (2.80, 2.84)
Non-Hispanic White	3.13 (2.43)	640	1.92 (1.92, 1.93)
Hispanic	3.10 (2.38)	603	2.71 (2.70, 2.72)
Multiracial	3.45 (2.57)	21	1.93 (1.80, 2.18)
Other	3.35 (2.57)	2	1.24 (0.55, 2.60)
Unknown	2.32 (1.97)	39	3.69 (3.51, 4.05)
Neighborhood Deprivation Index (NDI), n (%)			
Q1	3.27 (2.47)	280	1.76 (1.76, 1.78)
Q2	3.21 (2.45)	335	2.14 (2.13, 2.16)
Q3	3.02 (2.37)	369	2.51 (2.49, 2.53)
Q4	2.95 (2.34)	394	2.76 (2.75, 2.79)
Q5	3.14 (2.49)	390	2.57 (2.56, 2.59)
Smoking status, n (%)			
Current or Past	3.07 (2.44)	989	2.50 (2.49, 2.50)
Never smoked	3.17 (2.41)	779	2.16 (2.16, 2.17)
Comorbidities, n (%)			
Obesity	3.16 (2.39)	960	2.60 (2.60, 2.61)
Pre-diabetes	3.16 (2.40)	567	2.62 (2.61, 2.64)
Diabetes	3.04 (2.38)	648	2.36 (2.35, 2.37)
Hypertension	3.12 (2.43)	1106	2.18 (2.17, 2.18)
Chronic Kidney Disease, Stage 3–5	2.83 (2.34)	222	1.90 (1.89, 1.93)
Dyslipidemia	3.15 (2.42)	1085	2.25 (2.25, 2.26)
Cardiovascular Disease	3.12 (2.43)	1196	2.19 (2.19, 2.20)
Coronary Artery Disease	2.72 (2.30)	161	1.75 (1.74, 1.78)
Myocardial Infarction	2.68 (2.33)	67	1.50 (1.47, 1.55)
Peripheral Vascular Disease	2.78 (2.23)	383	2.01 (2.00, 2.02)
Cerebral Vascular Disease	2.73 (2.34)	115	1.71 (1.69, 1.75)
Transient Ischemic Attack	2.86 (2.39)	36	1.65 (1.59, 1.77)
Charlson Comorbidity Index			
0–1	3.25 (2.45)	953	2.57 (2.57, 2.58)
2	3.46 (2.48)	284	2.29 (2.28, 2.32)
3	3.20 (2.39)	178	2.38 (2.36, 2.43)
4+	2.70 (2.30)	353	1.88 (1.87, 1.89)
Year of incident adrenal adenoma diagnosis			
2015	5.28 (3.20)	55	0.41 (0.41, 0.42)
2016	4.87 (2.79)	61	0.47 (0.47, 0.48)
2017	4.30 (2.38)	98	0.78 (0.77, 0.79)
2018	3.67 (2.02)	130	1.17 (1.16, 1.19)
2019	2.07 (1.63)	190	1.94 (1.93, 1.97)
2020	2.34 (1.29)	300	4.47 (4.44, 4.54)
2021	1.67 (0.91)	448	7.75 (7.71, 7.85)
2022	0.98 (0.53)	486	13.65 (13.54, 13.85)

**Table 3 biomedicines-11-03167-t003:** Crude and adjusted hazard ratio (HR) and 95% confidence interval (95% CI) for the association between patient characteristics and completing a dexamethasone suppression test (DST) among all individuals with an adrenal adenoma, 2015–2022.

Patient Characteristics	HR (95% CI)	
Patient Demographics and Behavioral Characteristics	Crude, Bivariate Model	Fully Adjusted Model *
Age group		
18–39	Reference Group	Reference Group
40–49	1.10 (0.89, 1.38)	1.15 (0.92, 1.44)
50–59	0.95 (0.78, 1.17)	1.10 (0.89, 1.35)
60–69	0.89 (0.73, 1.08)	1.08 (0.87, 1.33)
70+	0.64 (0.52, 0.78)	0.84 (0.67, 1.05)
Sex		
Male	0.79 (0.72, 0.87)	0.84 (0.76, 0.93)
Female	Reference Group	Reference Group
Patient Race/Ethnicity		
Asian Pacific Islander	1.20 (1.00, 1.44)	1.11 (0.92, 1.34)
Black or African–American	1.46 (1.28, 1.67)	1.27 (1.10, 1.46)
Non-Hispanic White	Reference Group	Reference Group
Hispanic	1.40 (1.26, 1.57)	1.07 (0.95, 1.20)
Multiracial/Other/Unknown	1.31 (1.01, 1.71)	1.03 (0.79, 1.34)
Neighborhood Deprivation Index (NDI)		
Q1	Reference Group	Reference Group
Q2	1.21 (1.03, 1.42)	1.19 (1.01, 1.39)
Q3	1.36 (1.16, 1.58)	1.21 (1.03, 1.41)
Q4	1.47 (1.26, 1.71)	1.23 (1.05, 1.45)
Q5	1.43 (1.23, 1.67)	1.33 (1.13, 1.57)
Smoking status		
Current or former	0.85 (0.77, 0.93)	
Never	Reference	
Comorbidities		
Obesity	1.26 (1.15, 1.39)	1.08 (0.98, 1.20)
Prediabetes	1.19 (1.08, 1.32)	
Diabetes	0.99 (0.90, 1.09)	1.14 (1.02, 1.29)
Hypertension	0.82 (0.74, 0.90)	0.91 (0.82, 1.02)
Chronic kidney disease, stage 3–5	0.74 (0.64, 0.85)	
Dyslipidemia	0.92 (0.84, 1.01)	
Cardiovascular disease	0.81 (0.73, 0.90)	
Coronary artery disease	0.66 (0.56, 0.77)	
Myocardial Infarction	0.57 (0.45, 0.73)	
Peripheral Vascular Disease	0.75 (0.67, 0.84)	
Cerebral Vascular Disease	0.65 (0.54, 0.79)	
Stroke	0.60 (0.44, 0.82)	
Transient ischemic attack	0.67 (0.48, 0.93)	
Intracerebral hemorrhage		
Deep Vein Thrombosis	0.32 (0.15, 0.72)	
Osteoporosis	0.66 (0.56, 0.78)	
Osteopenia	0.85 (0.74, 0.98)	
Charlson Comorbidity Index		
0–1	Reference Group	Reference Group
2	0.92 (0.81, 1.06)	1.01 (0.87, 1.16)
3	0.92 (0.78, 1.08)	1.01 (0.85, 1.21)
4+	0.66 (0.58, 0.74)	0.80 (0.67, 0.96)
Year of incident adrenal adenoma diagnosis		
2015	Reference Group	Reference Group
2016	1.24 (0.85, 1.81)	1.53 (1.04, 2.25)
2017	2.14 (1.50, 3.04)	3.13 (2.12, 4.60)
2018	3.39 (2.40, 4.80)	5.62 (3.78, 8.63)
2019	5.63 (4.02, 7.88)	10.86 (7.01, 16.81)
2020	12.40 (8.93, 17.23)	26.71 (16.86, 42.34)
2021	18.23 (13.18, 25.02)	41.45 (25.65, 66.97)
2022	22.44 (16.23, 31.03)	54.03 (33.01, 88.43)

* Fully adjusted model was developed using Andersen’s behavioral model of health services conceptual framework. The fully adjusted model includes age, sex, race/ethnicity, obesity, diabetes, hypertension, the Charlson comorbidity index, and the year of incident adrenal adenoma diagnosis. Hazard Ratios and 95% Confidence Intervals are reported for the covariates included in the fully adjusted model.

**Table 4 biomedicines-11-03167-t004:** Patient demographics and clinical characteristics stratified by dexamethasone suppression test (DST) stratified by normal (DST ≤ 1.8 μg/dL) and elevated (DST > 1.8 μg/dL) values, overall and by site, 2015–2022.

	Total Population N (%)		KP Georgia N (%)		KP Southern California N (%)	
Patient Characteristics at Time of Dexamethasone Suppression Test (DST)	DST ≤ 1.8 μg/dL N = 1103 (62.4)	DST > 1.8 μg/dL N = 665 (37.6)	DST ≤ 1.8 μg/dL N = 138 (60.8)	DST > 1.8 μg/dL N = 89 (39.2)	DST ≤ 1.8 μg/dL N = 965 (62.6)	DST > 1.8 μg/dL N = 576 (37.4)
Age, mean (SD)	59.2 (13.0)	62.4 (12.2)	57.4 (10.8)	60.1 (10.8)	59.5 (13.2)	62.7 (12.3)
Age group, n (%)						
18–39	92 (8.5)	27 (4.1)	7 (5.1)	4 (4.5)	85 (8.8)	23 (4.0)
40–49	159 (14.4)	79 (11.9)	22 (15.9)	11 (12.4)	137 (14.2)	68 (11.8)
50–59	262 (23.8)	149 (22.4)	50 (36.2)	27 (30.3)	212 (22.0)	122 (21.2)
60–69	347 (31.5)	210 (31.6)	38 (27.5)	30 (33.7)	309 (32.2)	180 (31.3)
70+	243 (22.0)	200 (30.1)	21 (15.2)	17 (19.1)	222 (23.0)	183 (31.8)
Sex, n (%)						
Male	407 (36.9)	231 (34.7)	55 (39.9)	35 (39.3)	352 (36.5)	196 (34.0)
Female	696 (63.1)	434 (65.3)	83 (60.1)	54 (60.7)	613 (63.5)	380 (66.0)
Patient Race/Ethnicity, n (%)						
Asian Pacific Islander	90 (8.2)	49 (7.4)	3 (2.2)	4 (4.5)	87 (9.0)	45 (7.8)
Black or African–American	175 (15.9)	149 (22.4)	67 (48.6)	45 (50.6)	108 (11.2)	104 (18.1)
Non-Hispanic white	386 (35.0)	254 (38.2)	58 (42.0)	35 (39.3)	328 (34.0)	219 (38.0)
Hispanic	413 (37.4)	190 (28.6)	4 (2.9)	4 (4.5)	409 (42.4)	186 (32.3)
Multiracial	10 (0.9)	11 (1.7)	1 (0.8)	1 (1.1)	9 (0.9)	10 (1.7)
Other	1 (0.1)	1 (0.2)	0 (0.0)	0 (0)	1 (0.1)	1 (0.2)
Unknown	28 (2.5)	11 (1.7)	5 (3.6)	0 (0)	23 (2.4)	11 (1.9)
Neighborhood Deprivation Index (NDI), mean (SD)						
Q1	−0.93 (0.22)	−0.89 (0.24)	−1.04 (0.27)	−1.13 (0.29)	−0.92 (0.21)	−0.86 (0.22)
Q2	−0.39 (0.13)	−0.40 (0.14)	−0.46 (0.14)	−0.51 (0.12)	−0.39 (0.13)	−0.39 (0.13)
Q3	0.06 (0.15)	0.09 (0.14)	−0.06 (0.11)	−0.02 (0.10)	0.07 (0.15)	0.10 (0.14)
Q4	0.59 (0.20)	0.60 (0.22)	0.39 (0.15)	0.34 (0.17)	0.62 (0.19)	0.66 (0.19)
Q5	1.58 (0.55)	1.54 (0.52)	1.28 (0.84)	1.20 (0.62)	1.61 (0.49)	1.59 (0.49)
Smoking status, n (%)						
Current or Past	452 (41.0)	327 (49.2)	67 (48.5)	48 (53.9)	385 (39.9)	297 (51.6)
Never smoked	651 (59.0)	338 (50.8)	71 (51.5)	41 (46.1)	580 (60.1)	279 (48.4)
Lab Values, mean ± SD (median)						
HbA1c	6.3 ± 1.2 (5.9)	6.5 ± 1.4 (6.0)	6.3 ± 1.3 (5.8)	6.2 ± 1.3 (5.8)	6.3 ± 1.2 (6.0)	6.5 ± 1.4 (6.0)
Creatinine	0.9 ± 0.5 (0.9)	1.1 ± 0.9 (0.9)	0.9 ± 0.3 (0.9)	1.2 ± 0.7 (0.9)	0.9 ± 0.5 (0.8)	1.1 ± 1.0 (0.9)
Total cholesterol	177.2 ± 43.0 (177.0)	175.2 ± 45.9 (172)	187.9 ± 41.8 (186.0)	173.9 ± 41.3 (174)	175.8 ± 42.9 (176.0)	175.4 ± 46.6 (172.0)
LDL	106.5 ± 36.0 (105.0)	102.8 ± 39.5 (98.5)	120.1 ± 37.0 (117.0)	101.6 ± 32.3 (99.0)	103.9 ± 35.3 (102.0)	103.0 ± 40.8 (98.0)
HDL	50.1 ± 13.2 (49.0)	50.9 ± 15.3 (48.0)	51.1 ± 13.3 (50.0)	46.3 ± 13.1 (45.3)	50.0 ± 13.2 (48.0)	51.5 ± 15.5 (49.0)
Triglycerides	171.3 ± 110.5 (141.0)	178.8 ± 137.5 (142.0)	122.2 ± 78.4 (104.6)	148.8 ± 78.5 (134.0)	184.2 ± 114.2 (148.0)	183.6 ± 145.6 (151.0)
Potassium	4.0 ± 0.4 (4.0)	4.0 ± 0.5 (4.0)	4.0 ± 0.4 (4.0)	4.0 ± 0.4 (4.0)	4.0 ± 0.4 (4.0)	4.0 ± 0.5 (4.0)
Comorbidities, n (%)						
Obesity	626 (56.8)	334 (50.2)	84 (60.9)	53 (59.6)	542 (56.2)	281 (48.8)
Pre-diabetes	365 (33.1)	202 (30.4)	44 (31.9)	29 (32.6)	321 (33.3)	173 (30.0)
Diabetes	343 (31.1)	305 (45.9)	38 (27.5)	37 (41.6)	305 (31.6)	268 (46.5)
Hypertension	637 (57.0)	469 (70.5)	86 (62.3)	69 (77.5)	601 (62.3)	392 (68.1)
Chronic Kidney Disease, Stage 3–5	111 (10.1)	111 (16.7)	11 (8.0)	12 (13.5)	100 (10.4)	99 (17.2)
Dyslipidemia	656 (59.5)	429 (64.5)	55 (39.9)	37 (41.6)	582 (62.2)	388 (67.7)
Cardiovascular Disease	696 (63.1)	500 (75.2)	93 (67.4)	72 (80.9)	603 (62.5)	428 (74.3)
Coronary Artery Disease	91 (8.3)	70 (10.5)	7 (5.1)	10 (11.2)	84 (8.7)	60 (10.4)
Myocardial Infarction	37 (3.4)	30 (4.5)	3 (2.2)	3 (3.4)	34 (3.5)	27 (4.7)

**Table 5 biomedicines-11-03167-t005:** Crude odds ratio (OR) and 95% confidence interval (95% CI) for dexamethasone suppression test (DST) results and patient characteristics comparing normal DST (DST ≤ 1.8 μg/dL) to elevated DST (DST > 1.8 μg/dL), 2015–2022.

Patient Characteristics	Crude OR (95% CI)
	DST > 1.8 μg/dL
Patient demographics and behavioral characteristics	
Age group	
18–39	Reference Group
40–49	1.69 (1.02, 2.81)
50–59	1.94 (1.21, 3.11)
60–69	2.06 (1.30, 3.27)
70+	2.80 (1.76, 4.48)
Sex	
Male	0.91 (0.74, 1.11)
Female	Reference Group
Patient race/Ethnicity	
Asian Pacific Islander	0.83 (0.57, 1.54)
Black or African–American	1.29 (0.99, 1.70)
Non-Hispanic White	Reference
Hispanic	0.69 (0.55, 0.88)
Other	0.90 (0.52, 1.54)
Neighborhood Deprivation Index (NDI)	
Q1	Reference Group
Q2	1.23 (0.89, 1.71)
Q3	1.19 (0.86, 1.64)
Q4	1.14 (0.83, 1.57)
Q5	1.20 (0.87, 1.65)
Smoking status, Current or Former	1.39 (1.15, 1.69)
Comorbidities	
Obesity	0.77 (0.63, 0.93)
Pre-diabetes	0.89 (0.72,1.09)
Diabetes	1.88 (1.54, 2.29)
Hypertension	1.75 (1.43, 2.15)
Chronic Kidney Disease, Stage 3–5	1.79 (1.35, 2.38)
Dyslipidemia	1.24 (1.02, 1.51)
Cardiovascular Disease	1.77 (1.43, 2.20)
Coronary Artery Disease	1.31 (0.94, 1.82)
Myocardial Infarction	1.36 (0.83, 2.23)
Peripheral Vascular Disease	1.30 (1.04, 1.64)
Cerebral Vascular Disease	1.63 (1.11, 2.38)
Stroke	1.77 (0.95, 3.28)
Transient Ischemic Attack	2.37 (1.21, 4.62)
Deep Vein Thrombosis	8.35 (0.97, 71.61)
Osteoporosis	1.24 (0.88, 1.75)
Osteopenia	1.26 (0.95, 1.68)
Charlson Comorbidity Index	
0–1	Reference Group
2	1.43 (1.09, 1.90)
3	1.64 (1.18, 2.27)
4+	2.77 (2.16, 3.56)
Year of incident adrenal adenoma diagnosis	
2015	Reference Group
2016	1.66 (0.77, 3.57)
2017	2.06 (1.03, 4.13)
2018	1.86 (0.95, 3.62)
2019	1.46 (0.77, 2.77)
2020	1.64 (0.89, 3.04)
2021	1.21 (0.66, 2.21)
2022	1.07 (0.58, 1.95)

## Data Availability

The data underlying this article cannot be shared publicly due to the privacy of individuals and the integrated healthcare members who participated in the study. The derived data will be shared, on reasonable request, with the corresponding author.

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
