# Peer review of "Dexamethasone Suppression Testing in a Contemporary Cohort with Adrenal Incidentalomas in Two U.S. Integrated Healthcare Systems"

_biomedicines, 2023, doi:10.3390/biomedicines11123167_

Round 1
Reviewer 1 Report
Comments and Suggestions for Authors
Interesting paper researching on clinical work-up in adrenal incidentalomas with potential ACS
introduction - nicely written paragraph, clearly presenting the current knowledge and practice on incidentally found adrenal masses with endocrine secretion capabilities. - No remarks
Material and methods - systematically describes the process of data collection and analysis - No remarks
Results - nicely presented on the two keypoints of the study - parameters correlating with DST completion and variables connected with positive DST >1.8 µg/dL - no remarks
Discussion and conclusions - the authors elegantly implement and compare their results with the available literature. the man conclusion is that DST is vastly underused in adrenal incidentalomas - a trend that is changing in the last few years due to updated policy. The second main finding is that DST is positive (i.e. the incidentaloma is active and a source of ACS) in up to a one third of those patients. As authors correctly stated, the correlation of positive DST with the risk factors studied in the two population will need larger studies and emphasize few other inevitable limitations in such a heterogenous and diverse study group
Author Response
Thank you for your timely review of our original research article. We appreciate your careful review of our findings and your feedback. Based on your comments, we have revised our manuscript to improve the clarity of our reporting. Please see attachment for specific comments. Revisions within the manuscript are marked in red font. Specific replies to comments are below. Thank you for the opportunity to revise the manuscript and consider our original research for publication.

Reviewer 2 Report
Comments and Suggestions for Authors
Dear Author
The manuscript “Dexamethasone suppression testing in a contemporary cohort 2 with adrenal incidentalomas in two U.S. integrated healthcare 3 systems” is interesting. Big sample size (24,259 adults) increase the power of the study and Generalizability.
The manuscript is well written and has ethical code and committee.
There are some points that would be better to solve before publication:
1-The word “neighborhood characteristics” is not clear. What is the importance of “neighborhood characteristics” with Autonomous cortisol secretion (ACS) as a covariant?
2-For age it would be better to report the rang or report mean age+-SD
3-What is the exact mean of “reference” in table 5?
Author Response

(The authors gave the same response as above.)
